# Cryo-EM structures of orphan GPR21 signaling complexes

Xi Lin [1,8], Bo Chen[1,8], Yiran Wu [1,8], Yingqi Han[2], Ao Qi[2,3], Junyan Wang[4], Zhao Yang[4], Xiaohu Wei [5], Tingting Zhao[6,7], Lijie Wu[1], Xin Xie[2,3,5], Jinpeng Sun[4], Jie Zheng[2,3] ✉, Suwen Zhao [1,5] ✉ & Fei Xu [1,5] ✉

GPR21 is a class-A orphan G protein-coupled receptor (GPCR) and a potential therapeutic target for type 2 diabetes and other metabolic disorders. This receptor shows high basal activity in coupling to multiple G proteins in the absence of any known endogenous agonist or synthetic ligand. Here, we present the structures of ligand-free human GPR21 bound to heterotrimeric miniGs and miniG15 proteins, respectively. We identified an agonist-like motif in extracellular loop 2 (ECL2) that occupies the orthosteric pocket and promotes receptor activation. A side pocket that may be employed as a new ligand binding site was also uncovered. Remarkably, G protein binding is accommodated by a flexible cytoplasmic portion of transmembrane helix 6 (TM6) which adopts little or undetectable outward movement. These findings will enable the design of modulators for GPR21 for understanding its signal transduction and exploring opportunity for deorphanization.

The GPR21 is an orphan receptor, i.e., without an identified endogenous ligand. GPR21 shares 71% sequence identity with GPR52[1,2]. However, they show different tissue distribution and are pathologically related to different diseases[3–6]. GPR21 is expressed in almost all tissues with high expression in the brain and spleen[7]. It transduces signals primarily through coupling to Gs protein to activate adenylate cyclase[8], and coupling to Gq and G15/16 protein to activate mitogen-activated protein kinases (MAPKs)[9]. Recent studies have reported that GPR21 may regulate body weight and glucose metabolism as GPR21 knockout mice showed improved glucose tolerance and insulin sensitivity[5]. GPR21 was also found to play a key role in coordinating the proinflammatory activity of macrophages in accordance with obesity-induced insulin resistance[6] and its overexpression in vitro was observed to markedly attenuate insulin signaling[9]. Overall, these findings suggest that GPR21 could be a novel target for type 2 diabetes and other metabolic disorders.

An understanding of orphan GPCR structure and function could open the door to previously untapped drug targets and promote the development of tool ligands for both basic and therapeutic research[10,11]. Structures for orphan GPR52 receptor in ligand-free, ligand-bound, and G-protein coupled states have been recently reported, which shed light on the structural basis of an unusual self-activation mechanism and uncovered a novel side pocket[12], raising new questions in the orphan GPCR field: (1) whether other orphan receptors with high basal activity may have similar structural features including a built-in agonist-like motif; (2) structural determinants of ligand recognition. Since GPR21 was reported to exhibit high level of basal activity[8], whether it may employ a similar self-activation mechanism to that of GPR52 is worth investigating. Furthermore, sequence analysis pointed out a less conserved side-pocket between GPR52 and GPR21 implying that understanding ligand recognition will

[1]iHuman Institute, ShanghaiTech University, Pudong, Shanghai, China. [2]Shanghai Institute of Materia Medica, Chinese Academy of Sciences, Shanghai, China. [3]School of Pharmaceutical Science and Technology, Hangzhou Institute for Advanced Study, University of Chinese Academy of Sciences, Hangzhou, China. [4]Key Laboratory Experimental Teratology of the Ministry of Education and Department of Biochemistry and Molecular Biology, School of Basic Medical Sciences, Cheeloo College of Medicine, Shandong University, Jinan, Shandong, China. [5]School of Life Science and Technology, ShanghaiTech University, Shanghai, China. [6]School of Chinese Materia Medica, Nanjing University of Chinese Medicine, Nanjing, China. [7]CAS Key Laboratory of Receptor Research, National Center for Drug Screening, Shanghai Institute of Materia Medica, Chinese Academy of Sciences, Shanghai, China. [8]These authors contributed equally: Xi Lin, Bo Chen, Yiran Wu. ✉e-mail: jzheng@simm.ac.cn; zhaosw@shanghaitech.edu.cn; xufei@shanghaitech.edu.cn

aid the search for tool ligands for GPR21 for which no ligand has been reported. Here, we present cryo-electron microscopy (cryo-EM) structures of human GPR21 complexed with miniGs (mGs)[13] and miniG15 (mG15)[14,15] in the absence of a ligand, respectively. Together with G protein dissociation assay, mutagenesis analysis, HDX-MS analysis and molecular dynamics (MD) simulations, a self-activation model with distinct TM6 conformation for GPR21 is proposed and validated.

## Results

### Cryo-EM structures of GPR21·mGs and GPR21·mG15

In order to test whether GPR21 can signal through Gs and G15/16 as reported[8,9], and to obtain stable GPR21-G protein complex for structural investigation, we first performed bioluminescence resonance energy transfer (BRET) assays to measure the G protein heterotrimer dissociation upon activation[16,17] and found that GPR21 has a constitutive G15 and Gs activity (Fig. 1a, b). To improve the low surface expression of the wild type GPR21 protein, we used a fusion approach, with thermostabilized apocytochrome b562RIL (BRIL) attached to the receptor N terminus (Wild type GPR21 with BRIL, GPR21(wt)). The stable GPR21(wt)-G protein complex was formed by mixing purified GPR21(wt) with the mini-Gα (mGα), Gβγ and the camelid antibody Nb35[18] in vitro. Size-exclusion chromatography (SEC) and SDS-PAGE analysis reveal that purified GPR21(wt) protein can form a monodispersed complex with mGs or mG15 in the absence of an agonist (Supplementary Figs. 1, 2). We finally determined the cryo-EM structures of GPR21(wt)-mGs and GPR21(wt)-mG15 complexes with nominal global maps at 3.3 Å and 3.8 Å

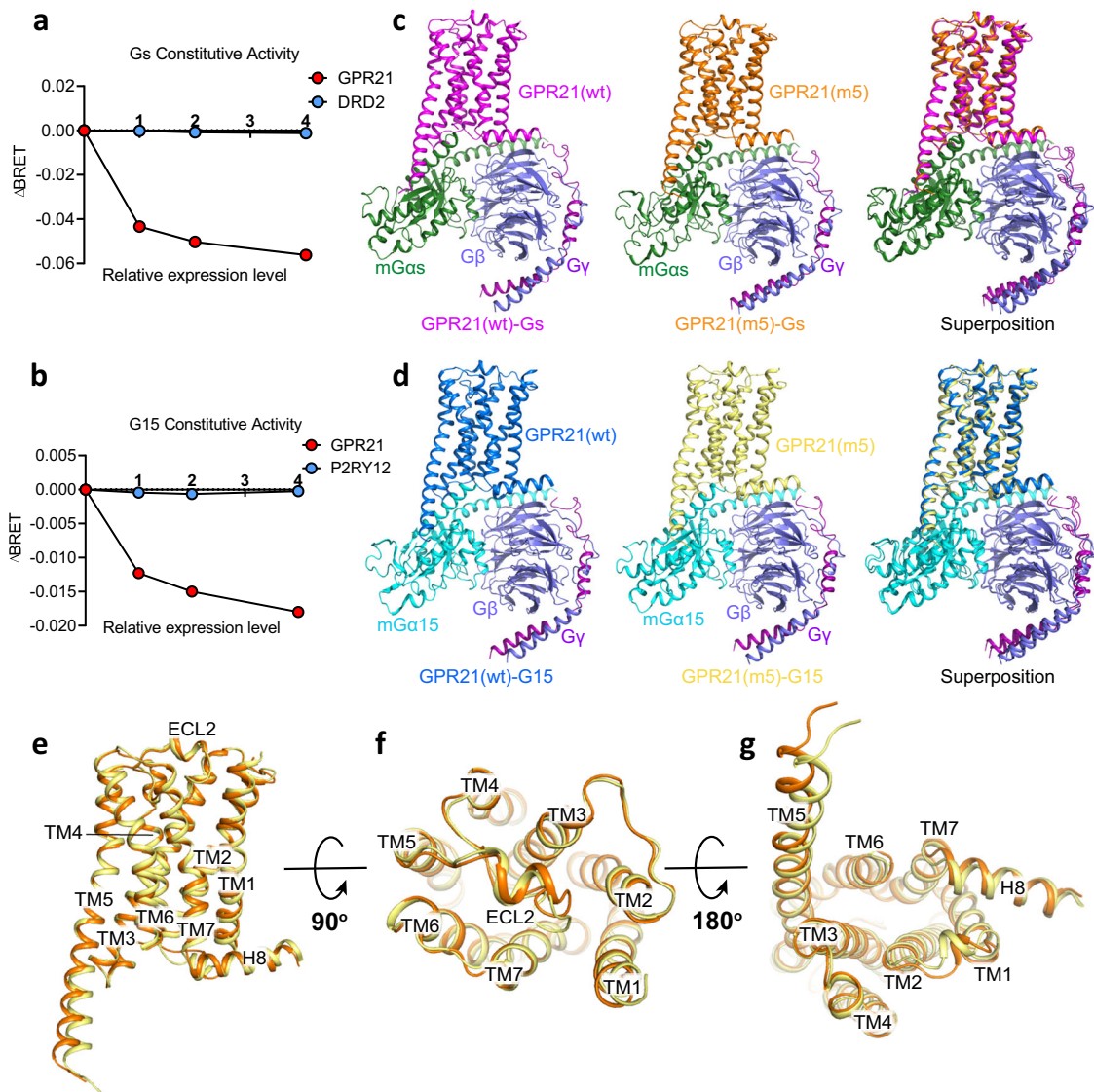

**Fig. 1 | Cryo-EM structures of the GPR21-Gs and GPR21-G15 complexes.**
**a** Constitutive activities of GPR21 receptor in Gs signaling measured by BRET assay (see "Methods"), dopamine receptor2 (DRD2) was used as a negative control (blue). The gradient cell surface expression levels of GPR21 receptor and DRD2 were achieved by adjusting the transfecting amounts of plasmids encoding the respective receptor in HEK293 cells. Data were from three independent experiments. ΔBRET: the change of bioluminescence resonance energy transfer value. Source data are provided as a Source Data file. **b** Constitutive activities of GPR21 receptor in G15 signaling measured by BRET assay (see "Methods"), purinergic Receptor P2Y (P2RY12) was used as a negative control (blue). Data were from three independent experiments. ΔBRET: the change of bioluminescence resonance energy transfer value. Source data are provided as a Source Data file. **c** Cryo-EM structures of the GPR21(wt)-mGs (left) and GPR21(m5)-mGs (middle). Superposition of the GPR21(wt)-mGs complex with the GPR21(m5)-mGs complex (right), wt, wild type; m5, 5mutations. **d** Cryo-EM structures of the GPR21(wt)-mG15 (left) and GPR21(m5)-mG15 (middle). Superposition of the GPR21(wt)-mG15 complex with the GPR21(m5)-mG15 complex (right), wt, wild type; m5, 5mutations. **e**–**g** Side (**e**), extracellular (**f**) and intracellular (**g**) views of the overlay between GPR21(m5)-mGs (orange) and GPR21(m5)-mG15 (yellow) structures. Transmembrane helices TM1-TM7 and helix 8 (H8) are labelled.

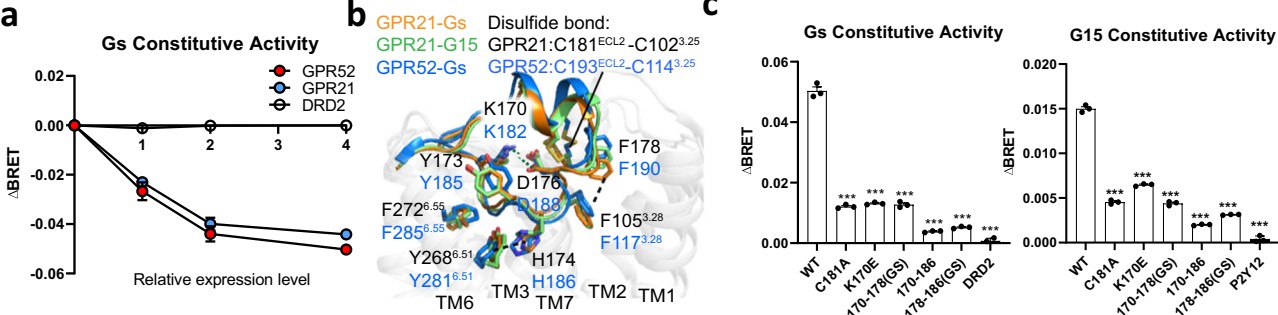

**Fig. 2 | Analysis of ECL2 in the orthosteric pocket. a** Comparison of constitutive activities of WT GPR21 and WT GPR52 in Gs signaling measured by BRET assay (see Methods). Data were from three independent experiments. DRD2 was used as a negative control (white). ΔBRET: the change of bioluminescence resonance energy transfer value. **b** Close-up view of the ECL2 in the orthosteric binding pocket between GPR21-Gs (orange), GPR21-G15 (green) and GPR52-Gs (blue, PDB: 6LI3). Key residues are shown as sticks. Salt bridges and π-π interactions are shown as green and black dashed lines, respectively. **c** Constitutive activities of WT GPR21 and GPR21 ECL2 mutants in Gs (left) and G15 (right) signaling measured by BRET assay (see Methods) (C181A and K170E, WT GPR21 with C181A mutation and K170E mutation, respectively; 170–186, WT GPR21 deleting residues 170–186; 170–178 (GS) and 178–186 (GS), residues 178–186 and 170–178, respectively, were replaced with a 6-residue linker (GGSGGS)). DRD2 and P2Y12 were used as a negative control for Gs and G15 signaling, respectively. ΔBRET: the change of bioluminescence resonance energy transfer value. Significance was determined by two-way analysis of variance (ANOVA) without repeated measures, followed by Dunnett's post hoc test (***$P < 0.001$). Data are mean ± s.e.m. ($n = 3$). Source data are provided as a Source Data file.

resolution, respectively. Some side chains were trimmed in the two models during model building since the map-model correspondence would be incomplete at some regions owing to the relatively low resolution (Supplementary Figs. 1, 2; Supplementary Table 1). In order to improve the resolution of the two complexes, 5 mutations (m5) are introduced: A118[3.41]W[19], C301[7.50]P, S305[7.54]A, N308[8.47]D, and V310[8.49]T; these mutations were designed according to the sequence homology of GPR21 to the GPR52 construct as previously reported[12,20] and were proven essential for high-resolution structure determination of the two complexes. We finally refined the cryo-EM structures of GPR21(m5)·mGs and GPR21(m5)·mG15 complexes with resolution both of 3.1 Å (Supplementary Figs. 3, 4). The overall structures of this mutant (GPR21(m5)-G protein) complexes largely overlap with those of the GPR21(wt)-G protein complexes with the receptor conformation nearly identical (root-mean-square deviation (r.m.s.d) between GPR21(m5) and GPR21(wt): 0.68 Å for Gs-bound and 0.90 Å for G15-bound structures) (Fig. 1c, d). We used the higher resolution structures of GPR21(m5) for structural illustration in the following analysis unless otherwise noted. The overall structure of GPR21 adopts a canonical seven-transmembrane (7TM) fold of class-A GPCRs (Fig. 1e–g).

## ECL2 contains a conserved agonist-like motif

We previously reported an agonist-like-motif (ALM) in ECL2 that constituted the structural basis for self-activation for GPR52[12] and we found that the basal activity of GPR21 was comparable to that of GPR52 (Fig. 2a). Here, we investigated the ECL2 conformation in two new structures in this study: GPR21-mGs and GPR21-mG15 complexes, and compared with that of GPR52-mGs complex.

Close examination of the GPR21-G protein structures revealed a conserved 22-residue ECL2 of GPR21 with a similar configuration to that of GPR52 (Fig. 2b). We found that the first half of GPR21's ECL2 (residues 170–178) was buried in the pocket while the second half (residues 179–186) was protruding into the extracellular surface. To maintain this unique configuration, the side chain of K170[ECL2] forms a key salt bridge with residue D176[ECL2], two π-π interactions (H174[ECL2]-Y268[6.51] and F178[ECL2]-F105[3.28]) form a local aromatic network, and a conserved disulfide bond is formed between C181[ECL2] and C102[3.25] in TM3, all of which are highly conserved in GPR52 (Fig. 2b). Mutations to disrupt the ECL2 conformation all dramatically reduced the basal activity of GPR21 (Fig. 2c; Supplementary Table 2). To further verify these results, we purified a construct of GPR21 protein with the entire

ECL2 deleted and tried to assemble with G protein in vitro. Consistent with the mutagenesis and cellular functional assays, purified GPR21 protein (on the template of GPR21(m5)) without ECL2 cannot form a GPCR-G protein complex (Supplementary Fig. 5). Taken together, these results confirm our hypothesis that the ECL2 functions as a built-in agonist to promote the high basal signaling activity for GPR21 and to stabilize the GPR21-G protein complexes.

## Structural basis of ligand recognition in the side pocket

While ECL2 occupies the canonical orthosteric pocket in GPR21, we were curious whether GPR21 exhibits a similar side pocket like what we have reported for GPR52[12]. To date there is no ligand available for GPR21, therefore, we looked into the possibility of GPR52 ligands (c17 and 7 m)[21,22] that may cross-react with GPR21. The side pocket, including residues from N-terminal and transmembrane regions, is only differed by eleven residues between the two receptors (Fig. 3a–c). However, our functional assay confirmed that WT GPR21 cannot be activated by c17 or 7 m (Fig. 3d)[12,21,22].

To identify key residues in ligand recognition between GPR21 and GPR52 in the side pocket and to guide screening of modulators for GPR21, we conducted mutagenesis and functional assays by mutating elements making up the side pocket of GPR21 into that of GPR52 (Fig. 3a–c). According to the structure and sequence alignment, we replaced the entire N-terminal region of GPR21 (M1-C28) with that of GPR52 (M1-C40), and mutated key interacting residues of GPR21 in the N-terminal region or in the transmembrane domain to corresponding residues in GPR52. Remarkably, GPR52's agonists can activate GPR21 signaling by increasing the cAMP accumulation when the entire N-terminal region or key residues in this region were replaced by corresponding residues in GPR52. However, mutations in the transmembrane domain did not yield any agonist binding activity. These results suggest that the N-terminal region is more essential in ligand recognition in the side pocket (Fig. 3d; Supplementary Table 3).

Next, to narrow down the ligand recognition code on the N-terminal region, we undertook back mutation one by one based on the GPR21 N-terminal mutants (Fig. 3e; Supplementary Table 3). We found that L16 in GPR21 (corresponding to P28 in GPR52) was the most important residue, since the GPR21 N-terminal mutants with the single back mutation of L16P could no longer be activated by GPR52's agonists (Fig. 3e; Supplementary Table 3). Therefore, this proline in GPR52 may be key for the N-terminal region to adopt the conformation required for binding to agonists at this site. Taken together, we

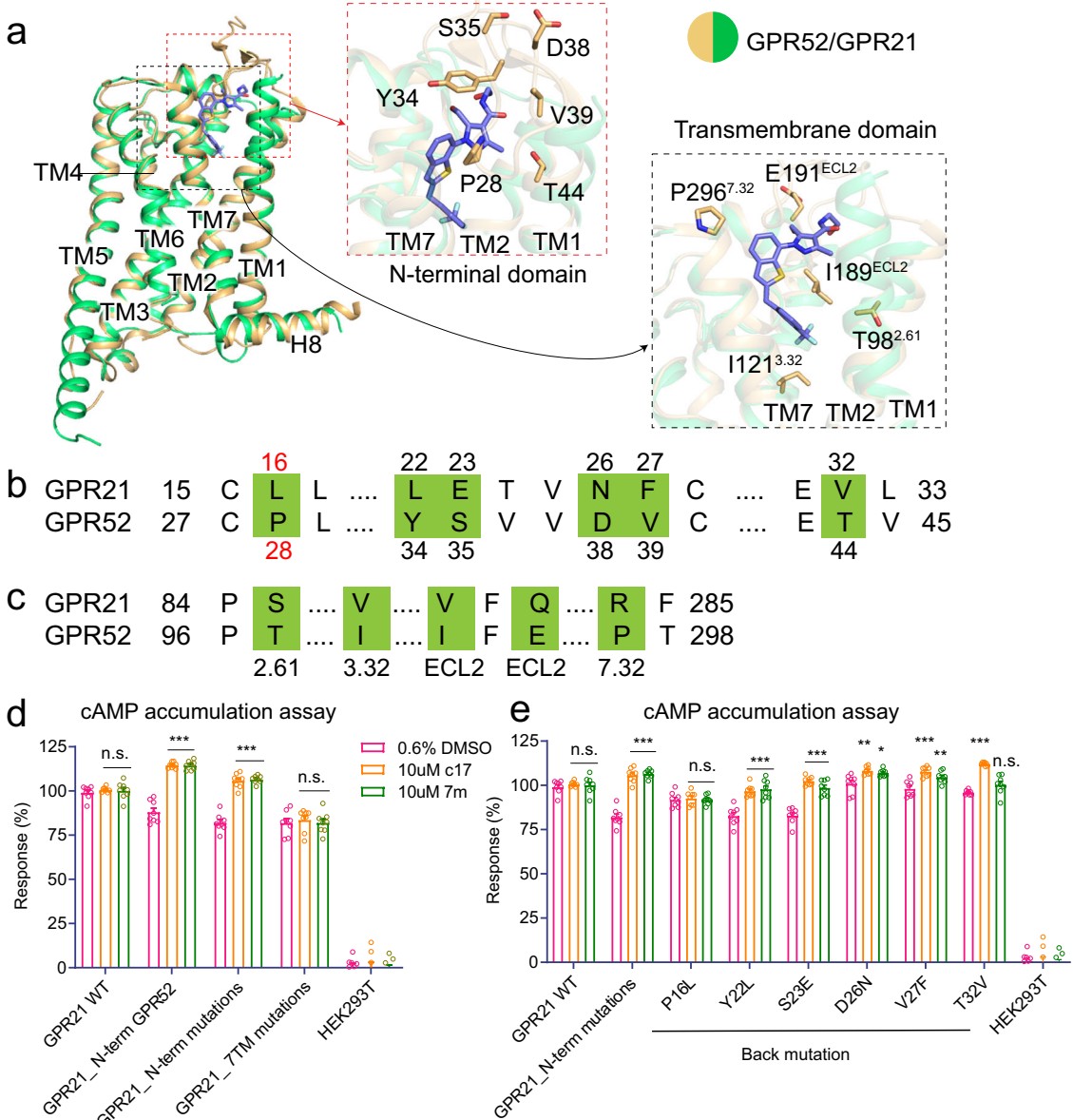

**Fig. 3 | Key residues for ligand recognition in the side pocket. a** Side and close-up views of the side pocket between GPR21 (green) and GPR52-c17 structure (gold). Key residues of N-terminal and transmembrane domains on GPR52 are shown as sticks. c17 (purple) is shown in the side pocket as sticks. **b, c** Sequence alignment of part of the N-terminal domain (**b**) and transmembrane domain (**c**) in GPR21 and GPR52. Key residues of side pocket are coloured in green. **d** Effects of GPR21 side pocket mutations on ligand binding of GPR52's agonists (c17 and 7 m) compared to DMSO-response at WT GPR21 receptor or mutants measured by cAMP accumulation assay (see Methods). Mutants include: replacing residues 1–28 of GPR21 with residues 1–40 of GPR52 based on WT GPR21 (GPR21_N-term GPR52); mutating 6 key residues (L16P, L22Y, E23S, N26D, F27V and T32V) of the N-terminal domain of GPR21 to that of GPR52 (GPR21_N-term mutations); and mutating 5 key residues (S86$^{2.61}$T, V109$^{3.32}$I, V177$^{ECL2}$I, Q179$^{ECL2}$E, and R283$^{7.32}$P) of the transmembrane domain of GPR21 to that of GPR52 (GPR21_7TM mutations). Data were normalized to surface expression. Significance was determined by two-way analysis of variance (ANOVA) without repeated measures, followed by Dunnett's post hoc test (***$P < 0.001$; NS not significance). Data are mean ± s.e.m ($n = 3$). **e** The effects of back mutations of GPR21_N-terminal mutants (in **b**) on the ligand binding of c17 and 7 m compared to DMSO-response (The method is the same as that of **d**). Significance was determined by two-way analysis of variance (ANOVA) without repeated measures, followed by Dunnett's post hoc test (***$P < 0.001$, **$P < 0.01$, *$P < 0.05$). Data are mean ± s.e.m. ($n = 3$). Source data are provided as a Source Data file.

concluded that the N-terminal region and especially the single residue L16 on GPR21 might be responsible for loss-of-binding for GPR52 ligand in the side pocket (Fig. 3b, e).

## Active conformation of GPR21

Having known that ligand-free GPR21 is capable of coupling to multiple G proteins to achieve self-activation through its built-in agonist-like motif in ECL2, next we analysed the active conformation of GPR21. Since there is no inactive-state GPR21 structure available, we employed an inactive model of GPR21 generated from GPCRDB[23] and alphafold[24], to compare with Gs-coupled GPR21 structure reported in this study.

Intriguingly, we found the overall receptor conformation of Gs-coupled GPR21 was similar to the inactive GPR21 (r.m.s.d for Cα is 0.55 Å) (Supplementary Fig. 6a, b). Given the high sequence homology and conserved ECL2 conformation between GPR21 and GPR52, we next compared the conformation of the Gs-bound GPR21 with the Gs-bound and Gs-free GPR52 structures, respectively. We found the overall receptor conformation of GPR21 was closer to GPR52 in the Gs-free state (r.m.s.d for Cα is 1.48 Å) than to GPR52 in the Gs-bound state (r.m.s.d for Cα is 1.79 Å). This analysis prompted that G-protein bound GPR21, in the absence of an agonist, may adopt a conformation highly resembling the inactive state. It is noteworthy the TM6 does not show

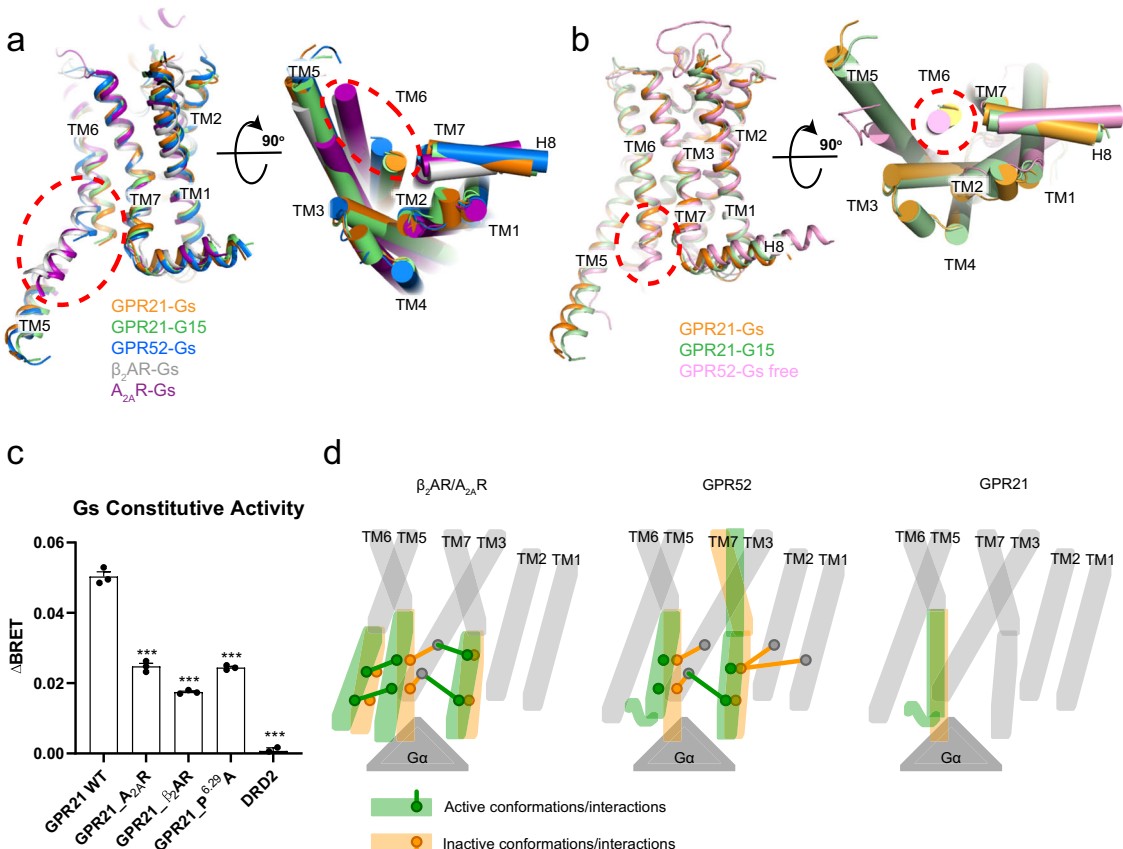

**Fig. 4 | Active conformation of GPR21. a** Superposition of GPR21-Gs, GPR21-G15, GPR52-Gs (PDB: 6LI3), $\beta_2$AR-Gs (PDB: 3SN6) and $A_{2A}$R-Gs (PDB: 5G53) structures. **b** Superposition of GPR21-Gs, GPR21-G15 and Gs-free GPR52 (PDB: 6LI1) structures. **c** Constitutive activities of WT GPR21 and GPR21 chimera receptors measured by BRET assay (see "Methods") (GPR21_$A_{2A}$R, replacing residues Q235$^{ICL3}$-V253$^{6.36}$ of WT GPR21 with residues E212$^{5.73}$-L241$^{6.43}$ of $A_{2A}$R. GPR21_$\beta_2$AR, replacing residues S233$^{ICL3}$-A251$^{6.34}$ of WT GPR21 with H241$^{ICL3}$-L272$^{6.34}$ of $\beta_2$AR. GPR21_P$^{6.29}$A, WT GPR21

with P$^{6.29}$A mutation) in Gs. DRD2 was used as a negative control. $\Delta$BRET: the change of bioluminescence resonance energy transfer value. Significance was determined by two-way analysis of variance (ANOVA) without repeated measures, followed by Dunnett's post hoc test (***$P < 0.001$). Data are mean ± s.e.m. ($n = 3$). Source data are provided as a Source Data file. **d** Working model of GPR21 in comparison to GPR52 and prototype class-A GPCRs.

any outward movement in G-protein coupled GPR21 (Fig. 4a, b). Meanwhile, other hallmarks of conformational change during class A GPCR activation[25], such as contact between R$^{3.50}$ in the DRY motif and Y$^{7.53}$ in the NPxxY motif[18], are not observed in the G-protein coupled GPR21 (Supplementary Fig. 6c, d).

The overall topology of the two types of G protein-bound GPR21 complex structures resemble each other, and the two G protein binding interfaces largely overlap on GPR21. We found the relative positions between GPR21 and G proteins were similar in the two complexes (Supplementary Fig. 7). Both structures share an overall conserved G protein binding interface mainly consisting of the TM3 and TM5–TM7 of GPR21 and the Gα subunit of the G protein hetero-trimers (Supplementary Fig. 7). Remarkably, an alpha helical extension of the cytoplasmic end of GPR21's TM5 covers a large area of the G protein-binding interface, resembling the conformation of TM5 in Gs-bound GPR52 and Gs-bound $\beta_2$ adrenergic receptor ($\beta_2$AR)[18] (Fig. 4a).

In comparison to the structures of Gs-bound $\beta_2$AR[18] and $A_{2A}$R[26], we found that the helical portion of TM6 in GPR21 and GPR52 became much shorter (Fig. 4a). Especially in GPR21, the cytoplasmic end (starting from A251$^{6.34}$) of TM6 cannot be observed in the electron density map likely owing to the high flexibility in this region. In order to understand the structural and functional properties of the ICL3 and the cytoplasmic end of TM6 (residues S233$^{ICL3}$-L254$^{6.37}$) (we named TM6-ICL3 region, Supplementary Fig. 8) that are disordered in the cryo-EM map, several GPR21 chimeras (based on the sequence alignment of GPR21, $A_{2A}$R, and $\beta_2$AR) were generated. Mutagenesis and

cellular functional assays showed that: replacing residues S233$^{ICL3}$-A251$^{6.34}$ of GPR21 with H241$^{ICL3}$-L272$^{6.34}$ of $\beta_2$AR (GPR21_$\beta_2$AR); replacing residues Q235$^{ICL3}$-V253$^{6.36}$ of GPR21 with residues E212$^{5.73}$-L241$^{6.43}$ of $A_{2A}$R (GPR21_$A_{2A}$R); or mutating the single kink residue P246$^{6.29}$ to alanine on ICL3 (GPR21_P$^{6.29}$A), all reduced the basal activity of GPR21 (Fig. 4c; Supplementary Table 4), suggesting that disruption of the TM6-ICL3 sequence may alter its native configuration thus impairing its G protein coupling activity.

In addition, we carried out HDX-MS experiments on GPR21 proteins[27] (constructs for HDX experiment were all based on GPR21(m5) template), GPR21_P$^{6.29}$A and GPR21_$\beta_2$AR chimeras, and compared them in the apo (ligand-free) state. Consistent with cellular functional assay (Fig. 4c), analysis of the HDX data showed that residues S233$^{ICL3}$-L254$^{6.37}$ of GPR21 and GPR21_P$^{6.29}$A adopt more heterogeneous conformations than that of GPR21_$\beta_2$AR, underlining unique dynamics mode of this region in solvents (Supplementary Fig. 9; Supplementary Table 5–7). To further investigate the conformations of the TM6-ICL3 region, we performed MD simulations of WT GPR21 (without G protein). Region S233$^{ICL3}$-A251$^{6.34}$ was modeled randomly in three conformations, and each was set as the starting point for three independent 2 μs runs. We analysed the radius of gyration of the cytoplasmic end of TM6, A244$^{6.27}$-A251$^{6.34}$ (disordered in GPR21), and found this region has different conformations with diverse degrees of folding in GPR21 (Supplementary Fig. 10). While for class A GPCRs with known functions, this region remained as α-helix in published MD simulations[28,29]. This analysis

indicated the local flexibility of the cytoplasmic end of TM6, in consistent with the results of functional data and HDX-MS experiments. By combining the HDX, MD and functional data, we tried to construct activation models for GPR21 in comparison to GPR52 and prototype class A GPCRs (Supplementary Fig. 11). The models show distinct features for the two highly similar orphan GPCRs when coupling to G proteins among which GPR21 adopts the least conformational changes (Supplementary Fig. 11).

## Discussion

Here, we determined the cryo-EM structures of the human GPR21 receptor coupled to two different G proteins in the absence of an agonist. A conserved ECL2 conformation in GPR21 was found essential for its high basal activity, which may provide the structural basis for understanding the self-activation mechanism for new orphan GPCRs. The HDX experiment and MD stimulation indicate that the cytoplasmic end of TM6 and ICL3 in GPR21 are quite flexible and may adopt multiple conformations, which may explain why we cannot observe the cryo-EM density map at this region. We hypothesize that such a non-canonical mechanism through flexible TM6 may provide GPR21 with the ability of coupling to multiple G proteins.

This research of GPR21 is, to our knowledge, the first time a class-A GPCR has not been observed a profound outward movement of TM6 and the repacking of the microswitches DRY and NPxxY when coupling to G protein. It prompted us to re-investigate the activation pathway for class-A GPCRs. In $\beta_2AR$ and $A_{2A}R$, which represent the common activation mechanism of class-A GPCRs, the inter-helical interaction map (Supplementary Fig. 11) show that TM6 moving apart from TM3 and repacking with TM5, and $Y^{7.53}$ getting close to TM3[25]. In the activation of GPR52, some of these transitions (TM6-TM3 moving apart and $Y^{7.53}$-TM3 getting close) remain, but most interactions in TM6-TM5 repacking are missing. The reason might be that GPR52 lacks the $P^{5.50}$-$I^{3.40}$-$F^{6.44}$ motif and thus has to adopt a different local conformational change during activation. Also, TM7 of GPR52 twists, as exhibited by movement of residues 7.45 and 7.46, which is not observed in $\beta_2AR^{18,30}$ or $A_{2A}R^{26,31}$ (Supplementary Fig. 11). As expected, such twist of TM7 is not observed in GPR21, for GPR21 resembles the conformation of Gs-free GPR52 structure.

In sharp contrast, the G protein-bound structure of GPR21 poses unusual inactive-like inter-helical interactions (Supplementary Fig. 11), suggesting its active conformation is distinct from either $\beta_2AR/A_{2A}R$ or GPR52. Correspondingly, the structure of active-sate GPR21 highly resembles Gs-free GPR52 in their overall conformation except for the uncoiling at the cytoplasmic end of TM6. For example, GPR21 also lacks the twist of TM7 in Gs-bound GPR52 structure; this might be due to more compact packing at the extracellular end, preventing the movement of TM7: there are two extra strong interacting pairs, a salt bridge between $E31^{1.35}$ and $R283^{7.32}$ and a π-π interaction of $F27^{1.31}$-$F284^{7.33}$ in GPR21 (Supplementary Fig. 12). Taken together we suggest models of GPR52 and GPR21 activation are different to common activation mechanism: both GPR21 and GPR52 undergo flexible TM6, when coupling to different G proteins, yet only GPR21 adopts an inactive-like conformation (Fig. 4d).

Orphan GPCR research is still in its infancy, and the research we have presented here is only the second orphan GPCR "self-activation" structural basis of G protein coupling to be reported. The lack of outward movement of a flexible TM6 when coupled to different G proteins, together with the inactive-like inter-helical conformation, shedding light on an active conformation distinct from all other known GPCRs. The unveiling of GPR52 and GPR21 self-activation and the ligand recognition on the two receptors will help to significantly advance research into signal transduction, drug discovery, and deorphanization of orphan GPCRs.

## Methods

### Protein expression of human GPR21

The gene of human GPR21-WT was synthesized by GenScript. The human *GPR21* gene (UniProt ID: Q99679) was subcloned into the expression vector pFastbac1. The C-terminal of GPR21 followed by an HRV 3 C protease recognition site and a 10x His tag. This construct was expressed with haemagglutinin signal peptide, Flag tag, and thermostabilized *Escherichia coli* apocytochrome *b*562RIL (BRIL) on the N-terminus to increase the expression yield (GPR21(wt)).

At the same time, to further improve expression yield and protein stability, we screened another construct, which has C-terminal truncations of the GPR21 residues 328–349, five point mutations introduced by site-directed mutagenesis: $A118^{3.41}W^{19}$, $C301^{7.50}P$, $S305^{7.54}A$, $N308^{8.47}D$, and $V310^{8.49}T$ (GPR21(m5)). These mutations refer to the previous literature to obtain GPR52 structure[12], originally designed by computational approaches (CompoMug)[32]. In addition, the mutant and chimera proteins used in this paper also contained these five mutations, including GPR21_$\beta_2AR$(m5) and GPR21_P6.29 A(m5).

About the expression of GPR21 protein, we used the Bac-to-Bac Baculovirus System in *Spodoptera frugiperda* (*Sf*9) cells (Invitrogen, 12659017) for expression[33]. These cells were infected at a density of $2 \times 10^6$ cells per mL with baculovirus. Cells were grown at 27 °C and collected at 48 h after infection, and cell pellets were stored at −80 °C for future use.

### GPR21 protein purification for complex formation

The cell pellets of GPR21 protein were thawed and washed with a low-salt buffer containing 10 mM HEPES pH 7.5, 20 mM KCl, 10 mM $MgCl_2$, protease inhibitor cocktail (Roche), and discard the supernatant by centrifugation at $38,000 \times g$ for 30 min. The cell pellets were followed by three washes with a high-salt buffer containing 10 mM HEPES pH 7.5, 1 M NaCl, 20 mM KCl, 10 mM $MgCl_2$ and protease inhibitor cocktail, and discard the supernatant by centrifugation at $38,000 \times g$ for 30 min each time. Before solubilization, purified cell pellets were resuspended and incubated with 2 mg ml$^{-1}$ iodoacetamide (Sigma) at 4 °C for 30 min. GPR21 was extracted from the membrane by adding HEPES, NaCl, lauryl maltose neopentyl glycol (LMNG) (Anatrace), and cholesteryl hemisuccinate (CHS, Sigma) to the membrane solution to a final concentration of 50 mM, 500 mM, 1.0% (w/v) and 0.2% (w/v), respectively, and stirred for 2 h at 4 °C. The supernatant was collected by centrifugation at $38,000 \times g$ for 30 min and incubated with TALON IMAC resin (Clontech) and 20 mM imidazole at 4 °C overnight. Then the resin was centrifuged at $800 \times g$ for 10 min and washed with 15 column volumes of buffer I containing 50 mM HEPES pH 7.5, 500 mM NaCl, 5% (v/v) glycerol, 0.05% (w/v) LMNG, 0.01% (w/v) CHS, 10 mM $MgCl_2$, 20 mM imidazole and followed by 15 column volumes of wash buffer II containing 25 mM HEPES pH 7.5, 100 mM NaCl, 5% (v/v) glycerol, 0.03% (w/v) LMNG, 0.006% (w/v) CHS, and 40 mM imidazole. Finally, the protein was eluted using 3 column volumes of elution buffer containing 25 mM HEPES pH 7.5, 100 mM NaCl, 5% (v/v) glycerol, 0.01% (w/v) LMNG, 0.0002% (w/v) CHS, and 220 mM imidazole. The protein solution was concentrated to ~3 mg ml$^{-1}$ for future use.

### Protein expression and purification of miniGαs, miniGα15, Gβ1γ2 and Nb35

The Gαs subunit of miniGs (miniGαs) used in this paper was the same as that used in the structures of the $A_{2A}R$-miniGs-Nb35 and GPR52-miniGs-Nb35[34,35]. miniGαs is composed solely of the GTPase domain from the adenylate cyclase stimulating G protein Gαs. In brief, miniGαs has three truncations based on wild-type Gαs: residues 1–5, residues 65–203, residues 255–264, and seven point mutations: G49D, E50N, A249D, S252D, I372A and V375I. The wild-type Gα15/16 (Synonyms:Gα15; UniProt ID: P30679) protein was particularly difficult to express and purify, so we produced a chimeric miniGα15 protein that is similar to previous studies[14,15]. In brief, the Gα15 was designed into a

multifunctional chimera based on miniGαs (same method as miniGs/q70[14]), the strategy was to transfer the specificity determinants (α5) of Gα15 onto miniGαs, providing possible binding sites for Nb35 antibody to stabilize the G protein heterotrimer for cryo-EM analysis[34].

About the purification of miniGα protein[35]. The miniGαs (or miniGα15) was cloned in pET15b vector and expressed in *E. coli* strain BL21 (DE3) (Thermo/Finnzymes, EC0114) and cultured in TB media supplemented with glucose (0.2%) and $MgSO_4$ (5 mM). Cultures were grown at 30 °C until an OD600 of 0.8 was reached. Expression was induced with IPTG (50 μM) and the temperature reduced to 25 °C. Cells were harvested 20 h postinduction by centrifugation at $2000 \times g$ for 20 min. The pellets from 1 L of *E. coli* culture were resuspended in buffer (40 mM HEPES pH 7.5, 100 mM NaCl, 10% glycerol, 10 mM imidazole, 5 mM $MgCl_2$, 100 μM GDP, 100 μg/ml lysozyme, 50 μg/ml DNase I, 100 μM DTT and protease inhibitor cocktail (Roche)) and lysed by sonication (10 min at 70% amplitude on ice). After another centrifugation ($40,000 \times g$, 30 min), the supernatant was loaded onto 2 ml $Ni^{2+}$ affinity chromatography. The column was washed with 30 ml of buffer (20 mM HEPES pH 7.5, 500 mM NaCl, 10% glycerol, 40 mM imidazole, 1 mM $MgCl_2$, 50 μM GDP). The column was eluted with 6 ml buffer (20 mM HEPES pH 7.5, 100 mM NaCl, 10% glycerol, 400 mM imidazole, 1 mM $MgCl_2$, 50 μM GDP). The protein solution was concentrated to a volume of 2 ml and loaded onto a Superdex200 10/600 column (GE) in buffer containing 10 mM HEPES pH 7.5, 100 mM NaCl, 10% glycerol, 1 mM $MgCl_2$, 10 μM GDP and 1 mM TCEP. Peak fractions of miniGα protein were concentrated to 20 mg ml⁻¹ for future use.

About the purification of heterodimeric Gβ1γ2 protein[35]. The heterodimeric Gβ1γ2 (human) was cloned into pFastbac-Dual vector and expressed in *Sf*9 for 48 h. The cell pellets from 2 L of Gβ1γ2 were thawed and resuspended to 50 ml in buffer (30 mM Tris pH 8.0, 100 mM NaCl, 5 mM $MgCl_2$, 5 mM imidazole, complete protease tablets (Roche), 50 μg/ml DNase I and 100 μM DTT). Cells were broken by sonication (10 min at 70% amplitude on ice) and clarified by centrifugation ($38,000 \times g$ for 1 h). The supernatant was loaded onto 2 ml $Ni^{2+}$ affinity chromatography. The column was washed with 20 ml of buffer (20 mM Tris pH 8.0, 300 mM NaCl, 30 mM imidazole, 10% glycerol and 1 mM $MgCl_2$). The column was eluted with 6 ml buffer (20 mM Tris pH 9.0, 50 mM NaCl, 500 mM imidazole, 10% glycerol and 1 mM $MgCl_2$). The elute was diluted to 60 ml in buffer (20 mM Tris pH 9.0, 50 mM NaCl, 10% glycerol, 1 mM $MgCl_2$, 1 mM DTT) and loaded onto a 5 ml Q FF column (GE Healthcare) at 5 ml/min. The Q FF column was washed with 40 ml buffer (20 mM Tris pH 9.0, 50 mM NaCl, 10% glycerol, 1 mM $MgCl_2$, 1 mM DTT) and eluted with a linear gradient of 50-300 mM NaCl in buffer (20 mM Tris pH 9.0, 50 mM NaCl, 10% glycerol, 1 mM $MgCl_2$, 1 mM DTT). The protein solution was concentrated to 1 ml and loaded onto a Superdex200 10/300 column (GE) in buffer (10 mM HEPES pH 7.5, 100 mM NaCl, 10% glycerol, 1 mM $MgCl_2$, 0.1 mM TCEP). Peak fractions of heterodimeric Gβ1γ2 protein were concentrated to 5 mg ml⁻¹ for future use.

About the purification of Nb35 protein[34,35]. The Nb35 was cloned in pET22b vector and expressed in *E. coli* strain BL21 (DE3) and cultured in TB media supplemented with glucose (0.2%) and $MgSO_4$ (5 mM). Cultures were grown at 30 °C until an OD600 of 0.8 was reached. Expression was induced with IPTG (50 μM) and the temperature reduced to 25 °C. Cells were harvested 20 h postinduction by centrifugation at $4000 \times g$ for 20 min. The pellet was resuspended from 1 L of *E. coli* culture in buffer (20 mM HEPES pH 7.5, 100 mM NaCl, 10 mM imidazole, 5 mM $MgCl_2$, Complete protease tablets, 50 μg/ml DNase I and 100 μg/ml lysozyme) and lysed by sonication (10 min at 70% amplitude on ice). After another centrifugation ($40,000 \times g$, 30 min), the supernatant was loaded onto 2 ml $Ni^{2+}$ affinity chromatography. The column was washed with 20 ml of buffer (20 mM HEPES pH 7.5, 500 mM NaCl, 40 mM imidazole). The column was eluted with 6 ml buffer (20 mM HEPES pH 7.5, 100 mM NaCl, 500 mM imidazole). The protein solution was concentrated to a volume of 1 ml and loaded onto a Superdex200 10/300 column (GE) in buffer (10 mM HEPES pH 7.5, 100 mM NaCl, 10% glycerol). Peak fractions of Nb35 protein were concentrated to 20 mg ml⁻¹ for future use.

## Complex formation for cryo-EM sample preparation

Purified GPR21 receptor, heterodimeric Gβ1γ2, miniGαs (or miniGα15), and Nb35 were mixed in a 1:1.2:1.5:2 molar ratio followed by the addition of apyrase (2 units), respectively. The mixture was incubated at 4 °C overnight. The GPR21-mGs (or GPR21-mG15) complex was loaded on size-exclusion chromatography (superdex 200 10/300 GL column, GE) in SEC buffer: 20 mM HEPES pH 7.5, 100 mM NaCl, 0.00075% (w/v) LMNG, 0.00025% GDN, 0.00025% (w/v) CHS, and 100 μM DTT. Peak fractions containing GPR21-G protein complex were concentrated to 1.5 mg ml⁻¹ for electron microscopy studies.

## Cryo-EM sample preparation and image acquisition

3 μl of the purified protein complexes (GPR21(wt)-mGs, GPR21(wt)-mG15, GPR21(m5)-mGs, GPR21(m5)-mG15,) were applied to glow-discharged 400-mesh Au grids (Quantifoil, R1.2/1.3) and subsequently vitrified using a FEI Vitrobot Mark IV at 100% humidity and 8 °C. All the datasets were collected on a Titan Krios 300 kV electron microscope (Thermo Fisher Scientifics, USA) equipped with a GIF Quantum energy filter (20 eV energy slit width, Gatan Inc., USA). The GPR21(m5)-mGs and GPR21(m5)-mG15 datasets were recorded by a K2 Summit direct electron detector (Gatan Inc, USA) at 130k nominal magnification (calibrated pixel size: 1.04 Å/pixel) and 8e⁻/pixel²/s; while the GPR21(wt)-mGs and GPR21(wt)-mG15 datasets were recorded by a K3 Summit direct electron detector (Gatan Inc, USA) at 105k nominal magnification (calibrated pixel size: 0.832 Å/pixel) and 15 e⁻/pixel²/s. The movies were recorded using the super resolution counting mode by SerialEM[36] which applies the beam image shift acquisition method with one image near the edge of each hole and saved as non-gain normalized TIFF files. A 50 μm C2 aperture was always inserted during the data collection period. The defocus ranged from −0.7 to −2.2 μm. For each movie stack, a total of 40 frames were recorded, yielding a total dose of 60e⁻/Å². For the GPR21(wt)-mGs, GPR21(wt)-mG15, GPR21(m5)-mGs and GPR21(m5)-mG15, a total of 4592, 4363, 3281 and 1353 movies were recorded, respectively.

## Cryo-EM image processing

All the datasets were motion corrected with MotionCor2[37] and no frame grouping. Both the dose weighted and non-dose weighted averages were saved, and the CTF parameters were estimated based on the non-dose-weighted averages using CTFFind[38]. Only images with the highest resolution of less than 4 Å were selected for further processing. Moreover, images with empty holes, visible contamination or large carbon regions by manual examination were also removed.

For the GPR21(m5)-mGs dataset, a total of 3133 movies were finally chosen for particle picking. To avoid potential bias about the structural conformation in the dataset, a Laplacian-of-Gaussian blob picker (Min:60 Å, max: 150 Å) in Relion3.1 was first applied to pick particles. 2D class averages with diverse orientations and clear secondary features were selected as the 2D templates for another round of autopicking process by Relion3.1, yielding an initial particle stack of 1,754,920 particles. Further rounds of 2D classification were applied to eliminate particles without visible secondary features by Relion 3.1[39], yielding a dataset containing 682,490 particles in total. These particles were imported into cryosparc 2.15[40] to generate five de novo initial models, and one model out of five, which 2D projection matched the majority of 2D class averages, was selected as the initial model for further processing. Subsequent 3D refinements, including heterogeneous 3D refinements, homogeneous refinements, and auto refinement of all the datasets, were

all performed by Relion3.1. For the first round 3D classification, this initial model was firstly low-pass filtered to 20 Å and used to divide the dataset into 3 different 3D classes. Masks and modification of the 3D maps were performed by the EMAN2 software[41]. 499,701 particles which were associated with 3D maps with GPCR complex features were grouped together and were subjective for further homogeneous refinement starting from the 40 Å low pass filtered initial map (search angle step size 7.5 degree). Custom python scripts (https://pan.baidu.com/s/10zTsJ_iVy0MSifBdYSoS7Q?pwd=rf3p) were also applied to eliminate 41,100 particles that could not converge to the consistent orientations (larger than 9 degrees) with higher than 50% chance among additional 7 3D refinement cycles after convergence. Finally, a total of 458,610 particles were selected for homogeneous refinement and post-processing, yielding a map with resolution of 3.3 Å determined by gold standard resolution test (cutoff: FSC = 0.143). Then, the dataset was subjected to further CTF refinement and Bayesian polishing and post-processing by Relion3.1, and the final resolution was improved to 3.12 Å with a soft-edge mask based on the 6 Å low pass filtered 3D refinement map with a 6 pixel extension and a 6 pixel soft edge (and the resolutions in subsequent cases were all determined with masks generated by this protocol). The local resolution was estimated using the cryosparc v2.15 "local resolution estimation" function (0.143 cut-off).

For the GPR21(m5)-mG15 dataset, the image processing followed the same scheme as the GPR21(m5)-mGs dataset. A total of 721,283 particles from 1226 recorded movies were boxed out and were subjected to further 2D classification by Relion3.1. Among them, 165,138 particles have clear secondary structural features. The GPR21(m5)-mGs refined map was used for the subsequent processes after low-pass filtered to 15 Å. After iterative 3D classifications by Relion 3.1, 119,479 particles were associated with 3D refined maps that had clear secondary structural features, and then these particles were grouped together and subjected to further rounds of 3D refinement by Relion3.1 starting from an 40 Å low pass filtered GPR21(m5)-mGs map. After CTF refinement and post-processing by Relion3.1, the resolution was improved to 3.12 Å.

For the GPR21(wt)−mGs dataset, the image processing followed the same scheme as the GPR21-mGs or GPR21-mG15 dataset. A total of 2,463,553 particles were boxed out and were subjective to further 2D classification processes by Relion 3.1. 830,312 particles were associated with the 2D averages with clear secondary features. The 15 Å low-pass filtered GPR21-mGs refined map was used as the initial model for subsequent 3D classifications by Relion 3.1. After iterative 3D classification, CTF refinement and/or Bayesian Polishing processes by Relion 3.1, 360,857 particles were grouped together and subjected to final 3D homogeneous refinements starting from the 40 Å low pass filtered GPR21(m5)-mGs map by Relion 3.1. The final resolutions determined by the gold standard test were determined as 3.27 Å.

For the GPR21(wt)−mG15 dataset, the raw movies were firstly motion corrected by Relion3.1. Then the motion corrected micrographs were imported into cryosparc v2.15.0 for patch CTF determination followed by blob picker particle picking. A total of 3,265,205 particles were boxed out and were subjective for further 2D classification processes by cryosparc v2.15.0. particles that display complex side view structures were grouped together and imported back into Relion 3.1 by pyEM v0.5 for subsequent rounds of 3D refinements. The 20 Å low-pass filtered GPR21-mGs refined map was used as the initial model for subsequent 3D classifications. After iterative 3D classification processes, 139,616 particles were grouped together and subjected to another round of final 3D refinement starting from the 40 Å low pass filtered GPR21(m5)-mGs map by Relion 3.1. After post-processing, the resolutions determined by the gold standard resolution test were 3.80 Å.

## Model building

The homology models of the GPCR, miniGs and miniG15 were initially generated by Swiss model (template: 6LI2, 7D3S)[12,42]. For Gβ, Gγ and Nb35, the model 3CIK, 6PCV and 6GDG were chosen[43–45]. These models were then fitted into the half1 density maps in UCSF Chimera[46], and manually adjusted to fit the density maps in Coot software[47]. Subsequently, the generated model was automatically refined and manually adjusted in Coot and Phenix[48], respectively, for several iterations. The clashscores, MolProbity, and Ramachandran analysis was performed using MolProbity[49]. The final refinement statistics were generated using the "comprehensive validation (cryo-EM)" function in Phenix. To avoid potential overfitting of the model, both FSCs between the model vs the half1 map (work) and half2 map (test) were determined (FSC = 0.5 cutoff). Structural figures were prepared in UCSF Chimera and PyMol.

## Measurement of receptor expression by ELISA

The wild-type (or mutant) GPR21 gene was subcloned in vector PTT5 with an N terminal haemagglutinin signal peptide and Flag tag. The GPR21 expression on cell surface was evaluated by the cell surface ELISA of Flag tag. HEK293 cells (ATCC, CRL-1573) were transiently transfected with varying concentrations of plasmid encoding target receptor, or co-transfected with Flag-tagged GPR21 together with G protein probes. Forty-eight hours after transfection, the cells were fixed with DPBS containing 4% formaldehyde for 5 min. Cells were washed 3 times in Tris-buffered saline containing Tween 20 (1:1000, TBST) and nonspecific binding sites were blocked by incubating cells for 1 h in blocking solution (5% BSA in TBST). Cells were washed 3 times and incubated for 1 h with Monoclonal ANTI-FLAG M2 antibody (1:1000, TBST; Sigma-Aldrich, F1804). After 3 washes in TBST, the cells were then incubated for 1 h with HRP-conjugated goat anti-mouse antibody (1:3000, TBST; Invitrogen, A-21235). The cells were further incubated with TMB/E solution (EMD Millipore, Billenca, MA, USA) and 0.25 M HCl was subsequently added to stop the reaction. The HRP activity was determined by measuring the absorbance at 450 nm using an Infinite M200 PRO microplate reader (Tecan, Männedorf, Switzerland)[50].

## Bioluminescence resonance energy transfer (BRET) assay

Gs BRET probe (Gαs(123)-Rlu8, Gβ and Gγ-GFP2) and G15 BRET probe (Gα15(245)-Rlu8, Gβ and Gγ-GFP2) were generated according to previous report (Gs BRET probe: Gαs(123)-Rlu8, Gβ and Gγ-GFP2. Gαs(123): Amino acid (AA) position number is 123, Amino acid (AA) position number indicates the position in the Gα protein of the first amino acid of the linker flanking the RLuc8 sequence; Gγ-GFP2: GFP2 was fused to the C-terminal of Gγ. G15 BRET probe: Gα15(245)-Rlu8, Gβ and Gγ-GFP2. Gα15(245): Amino acid (AA) position number is 245)[16]. The G protein dissociation assay was performed as previously described[17]. HEK293 cells were transiently co-transfected with varying amounts of plasmids encoding WT or mutated GPR21 together with G15 or Gs BRET probes. P2RY12 and DRD2 were used as control for G15 and Gs BRET assay, respectively. 24 h after transfection, cells were distributed into a 96-well microplate and incubated for additional 24 h at 37 °C. For the constitutive activity measurement, the transfected cells were washed twice with HBSS and the BRET signal was directly recorded using a Mithras LB940 microplate reader (Berthold Technologies) after the addition of coelenterazine 400a at a final concentration of 5 µM. The BRET signal was calculated as the ratio of light emission at 510 nm and light emission at 400 nm. ΔBRET represent the change of bioluminescence resonance energy transfer value. ΔBRET = BRET signal (GPCR-G protein sensor) - BRET signal (only G protein sensor).

## Cyclic AMP accumulation assay

The wild-type GPR21 gene was subcloned in vector PTT5 with an N terminal haemagglutinin signal peptide and Flag tag. Mutant GPR21 DNA were produced by QuickChange PCR. HEK293T cells (ATCC, CRL-11268) were cultured in $1 \times$ DMEM supplemented with 10% (v/v) fetal bovine serum (FBS) on 6-well cell-culture plates in 5% $CO_2$ at 37 °C. When cells had grown to around $2 \times 10^6$, 500 ng wild-type or mutant GPR21 DNA was transfected into cells by incubation with 1.5 µl Lipo-fectamine 2000 reagent (Life Technologies). Assays were started 24 h after transfection, cells were resuspended in 1X DMEM by 0.5 ml 0.05% trypsin at a density of $2 \times 10^5$ cells per ml and were plated in a 384-well assay plate (1000/well). Another 5 µl of DPBS buffer containing 10 µM c17 (or 7 m; or 0.6% DMSO) was added to the cells, then incubated at 37 °C for 30 min. Next, Plates were developed by adding 5 µl anti-cAMP and 5 µl cAMP-d2 antibody of work concentration, and they were incubated for 1 h at room temperature. Intracellular cAMP measurement was acquired using a Perkin-Elmer EnVision plate reader according to the manufacturer's instructions (Cisbio HTRF Dynamic 2 cAMP kit). The HTRF signal (HTRF ratio) was calculated as the ratio of light emission at 665 nm and light emission at 620 nm (665 nm/620 nm).

The HTRF ratio was converted to a response (%) using the following formula: response (%) = (RATIOsample – RATIOmean$_{HEK293T}$)/(RATIOmean$_{GPR21WT}$ – RATIOmean$_{HEK293T}$) × 100. RATIOsample: HTRF ratio of each sample; RATIOmean$_{GPR21WT}$: HTRF ratio of wild-type GPR21/HEK293T (mean), defined as 100% response; RATIOmean$_{HEK293T}$: HTRF ratio of HEK293T cells (mean), defined as 0% response. Data were analyzed using GraphPad Prism 7.0. Cell-surface expression for each receptor (wild-type receptor or mutants) was monitored by a fluorescence-activated cell sorting (FACS) assay. In brief, the expressed cells (10 µL) were incubated with 10 µL mouse anti-Flag M2–fluorescein isothiocyanate (FITC) antibody (1:100 in PBS; Sigma, F4049) for 20 min at 4 °C, and then a 9-fold excess of PBS was added to cells. Finally, the surface expression (2000 cells) of GPR21 or mutants was monitored by detecting the fluorescent intensity of FITC using a Guava EasyCyte HT system (Millipore).

## Molecular dynamics simulations

Structure of GPR21(m5)-Gs was used for the MD simulations. G protein and antibody were removed. Processing of the GPR21 structure was performed with the Protein Preparation Wizard tool[51] in Schrödinger Suite 2019-2. The missing region in ICL3 was generated in three different conformations using the Prime tool[52], and all residues were mutated back to wild type using the same tool. Molecular dynamics simulations were performed with GROMACS 2018.5[53] using the force field CHARMM36[54], which has been widely used in simulations of membrane proteins. The protonation states of residues were assigned automatically by the program based on their environment. As a result, all the lysine residues were protonated; all the histidine residues were neutral (either Nδ1 or Nε2 protonated); all the aspartate and glutamate residues except for Asp75$^{2.50}$ were not protonated. GPR21 were embedded into a pre-equilibrated POPC (1-palmitoyl-2-oleoyl-sn-glycero-3-phosphatidylcholine) lipid bilayer with TIP3P water (cubic box, $72.636 \times 72.636 \times 101.308$ Å$^3$) using the membed tool in the program GROMACS. The orientation of GPR21 in the membrane was determined by superposing to Gs-bound β$_2$AR in OPM database[55]. Sodium ions were added to a concentration of 0.15 M (concentration of Na$^+$ in blood and interstitial fluids) in water, and chloride ions were added to neutralize the system. The constructed systems contain ~54,000 atoms, including 125-127 POPC molecules. For each of the three starting points (with different ICL3 conformations), molecular dynamics simulations were run for three times independently. First, atom velocity was generated randomly for each independent run at a temperature of 310 K. Then the system was relaxed in a canonical

(NVT) ensemble (with Berendsen thermostat) for 300 ps and balanced in position-restrained molecular dynamics (isothermal–isobaric (NPT) ensemble with pressure of 1 atm, using semi-isotropic coupling, with Berendsen thermostat and Berendsen barostat) for 15 ns (total energy was stable). Finally, productive molecular dynamics with no position restraints (with Nose-Hoover thermostat and Parrinello-Rahman barostat) was run for 2 µs. Time steps used in the simulations are: 1 fs in the equilibrium steps, and 2 fs in the productive simulation. Trajectories of MD simulations have been deposited on BSM-Arc[56] and are available at https://doi.org/10.51093/bsm-00037.

## Hydrogen-deuterium exchange (HDX) detected by mass spectrometry (MS)

Peptide identification: Peptides were identified using tandem MS (MS/MS) with a Fusion Orbitrap mass spectrometer (ThermoFisher). Product ion spectra were acquired in data-dependent mode with the top eight most abundant ions selected for the product ion analysis per scan event. The MS/MS data files were submitted to Proteome Discover 2.4 (ThermoFisher) for high confident peptide identification.

HDX-MS analysis: 5 µM of GPR21, GPR21_P6.29 A, and GPR21-β$_2$AR (50 mM HEPES, pH 7.4, 150 mM NaCl, 5% glycerol, 5 mM MgCl$_2$, and 2 mM DTT) were incubated with and without the compound at molar ratio for 0.5 h (protein: ligand) before the HDX reactions at 4 °C. 4 µl of protein/protein complex with ligand/peptide was diluted into 16 µl D$_2$O on exchange buffer (50 mM HEPES, pH 7.4, 50 mM NaCl, and 2 mM DTT) and incubated for various HDX time points (e.g., 0, 10, 60, 300, 900 s) at 4 °C and quenched by mixing with 20 µl of ice-cold 1 M TCEP 100 mM NaH$_2$PO$_4$ (pH 2.5). A fully deuterated control was incubated in D2O buffer (50 mM HEPES, pH 8, 50 mM NaCl, 5 M guanidine hydrochloride and 2 mM DTT) overnight at room temperature. Each quenched sample was immediately injected into the LEAP Pal 3.0 HDX platform. Upon injection, samples were passed through an immobilized pepsin column (2 mm × 2 cm) at 120 µl min−1 and the digested peptides were captured on a C18 PepMap300 trap column (ThermoFisher) and desalted. Peptides were separated across a 2.1 mm × 5 cm C18 separating column (1.9 µm Hypersil Gold, ThermoFisher) with a linear gradient of 4–40% CH$_3$CN and 0.3% formic acid, over 6 min. Sample handling, protein digestion and peptide separation were conducted at 4 °C. Mass spectrometric data were acquired using a Fusion Orbitrap mass spectrometer (ThermoFisher) with a measured resolving power of 65,000 at m/z 400. HDX analyses were performed in duplicate or triplicate, with single preparations of each protein state. The intensity weighted mean m/z centroid value of each peptide envelope was calculated and subsequently converted into a percentage of deuterium incorporation. Statistical significance for the differential HDX data was determined by an unpaired $t$-test for each time point, a procedure that is integrated into the HDX Workbench software[57]. Corrections for back-exchange were automatedly processed by including the fully deuterated control during the HDX Workbench analysis. The bimodal distributions following EX1 exchange pattern (GPR21:SSQSGETGEVQACPDK-RYAMVL;GPR21(m5)_P6.29 A:SSQSGETGEVQACADKRYAMVL) are specifically analysed by the HX express2 software[58,59].

## Construction of inter-helical interaction maps

Receptors in active and inactive states used in analysis were: β$_2$AR, 3SN6[18] and 2RH1[30]; A$_2$AR, 5G53[26] and 4EIY[60]; GPR52, 6LI3 and 6LI1[12]. Active state-unique interaction is defined to two residues in different helices when: the distance between closest atoms is <4.6 Å in active state and >6.0 Å in inactive state. Correspondingly, inactive state-unique interaction is defined when: the distance between closest atoms is <4.6 Å in inactive state and >6.0 Å in active state. If a

pair of residues has state-unique interaction in $\beta_2AR$, $A_{2A}R$, or GPR52, the residues in GPR21 will be checked for whether they have interactions (distance of closest atoms is <4.6 Å). Structurally equivalent residues in different receptors are in consistent with the Ballesteros-Weinstein numbering (BWN)[61] except for TM5 where GPR21 and GPR52 have one-residue slide in structure comparing to BWN in class A receptors[12].

### Reporting summary

Further information on research design is available in the Nature Portfolio Reporting Summary linked to this article.

## Data availability

Data supporting the findings of this manuscript are available from the corresponding authors upon reasonable request. The coordinates for GPR21(wt)-mGs, GPR21(wt)-mG15, GPR21(m5)-mGs and GPR21(m5)-mG15 have been deposited in the Protein Data Bank with the accession codes 8HJ1, 8HJ2, 8HIX and 8HJ0. The EM maps for GPR21(wt)-mGs, GPR21(wt)-mG15, GPR21(m5)-mGs and GPR21(m5)-mG15 have been deposited in EMDB with the codes: EMD-33483, EMD-33481, EMD-33480 and EMD-33482, respectively. The HDX-MS raw data generated in this study have been deposited in a public repository under accession code IPX0005456000. The trajectories of MD simulations have been deposited in BSM-Arc (https://bsma.pdbj.org/) with the accession BSM00037 and are available at https://doi.org/10.51093/bsm-00037 (https://bsma.pdbj.org/entry/37). The custom python scripts for cryo-EM data processing are available (https://pan.baidu.com/s/10zTsJ_iVy0MSifBdYSoS7Q?pwd=rf3p). Source data are provided with this paper.

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

## Acknowledgements

This work was supported by the Ministry of Science and Technology of China (2018YFA0507000 to F.X. and S.Z.), National Natural Science Foundation of China (32071194 to F.X., 32122024, 31971178 to S.Z., and 22107072 to Y.W.), China Postdoctoral Science Foundation (BX20200349 to X.L.). We also thank the support from Shanghai Frontiers Science Center for Biomacromolecules and Precision Medicine at ShanghaiTech University. The cryo-EM data were collected at the Bio-Electron Microscopy Facility of ShanghaiTech University with the assistance of Q. Sun, D. Liu, L. wang and Z. Zhang. We thank J. Liu, N. Chen, Z. Fan and L. Jiang from Cell Expression Core, as well as Q. Tan, X. Liu, Q. Shi, S. Hu, F. Li, F. Zhou, L. Wang, P. Si and L. Zhang from Protein Purification, Assay and Cloning Cores of iHuman Institute for their support.

## Author contributions

X.L. performed cloning, purified the GPR21-G protein complex, prepared cryo-EM samples, collected EM data, performed cAMP functional assays of pocket mutants and structural analysis; B.C. collected EM data, determined structures, performed model building and refinement; Y.W. performed molecular dynamics simulations, constructed inter-helical interaction map, and structural analysis; Y.H. and A.Q. performed HDX-MS assays; Z.Y. and J.W. performed BRET assay; X.W. and T.Z. performed cell-based functional assays; L.W. assisted with model building; X.X. supervised functional assays; J.S. supervised the BRET assay and discussed the results; J.Z. supervised HDX-MS assays; S.Z. supervised molecular dynamics simulations, inter-helical interaction map construction, and structural analysis; F.X. conceived the project and designed and supervised all experiments. All authors contributed to data interpretation and preparation of the manuscript.

## Competing interests

The authors declare no competing interests.
