## [Peer Review File · Nature Communications]

REVIEWER COMMENTS

Reviewer #1 (Remarks to the Author):

Xi Lin et al. report the active structures of GPR21, which is one of the class A GPCRs, in complex with heterotrimeric miniGs and miniG15 proteins, respectively. They found an agonist-like motif in extracellular loop 2 that occupies the orthosteric pocket and promotes receptor activation. They also found that the flexible cytoplasmic portion of transmembrane helix 6 (TM6) adopts little or undetectable outward movement, which is different from other GPCRs such as β -2AR. In addition, they carried out HDX-MS experiments and molecular dynamics (MD) simulations to analyze the dynamic properties of TM6 and its neighboring residues. They executed their MD simulations of GPR21 without the G proteins, starting from three different modeled structures that modeled the missing residues of their EM structure. Their simulation length of 2 μ s and the number of parallel trajectories per system (three independent runs per modeled structure) are enough. Their simulation data exhibited local flexibility of TM6 and its neighbors, which agrees with the results of their functional and HDX-MS experiments. Therefore, their MD data could be published without any further revisions. Finally, since the structure of GPR21, which could be a potential novel drug target for diabetes, has not been reported before, their work would be worthwhile for publication in Nature Communications. However, there are some minor issues that should be revised in their manuscript before publication.

Minor issues:

(i) I think if there was a sequence alignment comparing multiple GPCRs (beta2-AR etc.), especially TM6 and its neighbors, their results would become more understandable and convincing.

(ii) Upon paper acceptance, they could submit their MD data to an archive, such as GPCRmd (<https://submission.gpcrmd.org/home/>) or BSM-Arc (<https://bsma.pdbj.org/>). Considering the commitment of this journal to sharing data associated with its original publications, the authors should consider doing so.

(iii) There are some typos such as 'A side pocket that may be employed ...' and 'induced-insulin resistance6. and its ...', so they should carefully revise their manuscript.

Reviewer #2 (Remarks to the Author):

The work presented by Professor Xu and colleagues is very interesting and adds value to the field. The experiments are well planned and executed, and the results are explained clearly.

I do have a few questions and suggestions for the authors:

1. The H/DX-MS data for TM6-ICL3 of all the three states of the protein is nicely depicted in supplementary figure 7. Interestingly, the authors showed that replacing Q235-V253 of apo-GPR21 (m5) with E212-L241 leads to the complete disappearance of the slowly exchanging population. I request the authors to provide the relative ratio between the two populations (Left and Right) for apo-GPR21 (m5) and apo-GPR21(m5)_P6.29A and the SD.
2. The legend of the Supp. Fig. 7a says mass spectra of 0s and 60s are shown, but there are also 10s, 300s, and 900s. Please rectify that.
3. Are the mean centroid values for the two populations of apo-GPR21 and apo-GPR21(m5)_P6.29A different? Please provide a table with the centroid values of each

population and SD.

4. The authors have not explained in their methods how the carryover effect was minimized. Could the bimodal distribution be because of the carryover effect?

5. I understand that the average back exchange for H/DX-MS experiments can range between 25-30%, and based on that, the authors have corrected the uptake in the HDX workbench software. But, the BE % varies from peptide to peptide based on the length and accessibility. Therefore, I request the authors to perform a fully deuterated control experiment to show that the particular region TM6-ICL3 does shift to the maximally exchanged population with time.

6. Are there any other overlapping peptides from this region? It would be worth looking at those since the authors hypothesize that P6.29 plays an important role.

7. The authors explain data rendering in their methodology, but there are no results showing this. Since the main data (peptide from TM6-ICL3) is showing a bimodal distribution, this would not be accurate. I suggest the authors remove this paragraph from their methodology.

I recommend the publication of the manuscript once the above concerns are satisfied.

Reviewer #3 (Remarks to the Author):

This is an interesting and solid paper which provides the first structures of the enigmatic orphan GPCR GPR21. Given the accelerating interest in orphan and understudied GPCRs this work will be important for a large number of biochemists, pharmacologists and cell biologists.

SPECIFIC COMMENTS:

1. The lower resolution structures are incomplete per my examination of the maps. A careful examination of the G15 structure, for instance, disclosed a large number of residues whose side-chains were not easily discernable in the maps. As well a number of discontinuities in the low resolution maps was evident. By contrast the best map (which is the one discussed in the paper) was fine and the model accurately fits the map. I would recommend the authors either delete the lower resolution maps or clearly state in the results and discussion that the map-model correspondence is incomplete and indicate in a separate figure where side-chains and backbone carbons have been modeled in without correspondence with the map.

2. For the estimates of constitutive activity the authors should as a control also have results with a GPCR with low levels of constitutive activity for comparison.

3. Given the authors do not have an inactive state structure to compare with the active state structures, the section on mechanisms of activation is premature and should be eliminated or shortened and also the authors need to indicate that they are not making these inferences based on any real knowledge of the inactive  active state transitions for GPR21

Reviewer #4 (Remarks to the Author):

In this manuscript, the authors investigated and compared the structures and dynamics of GPR21 and its mutant. The authors applied single particle cryo-EM, HDX, and other experiments to report the role of TM6 and a conserved ECL2 conformation in GPR21, essential for their high basal activity. The application of MD simulation also predicted the

dynamic nature of TM6. This work is important in the field of drug development and represents a good correlation of experimental results with theoretical predictions. However, when I went through it in detail, I found that it needs a lot of explanation to be a good fit for this journal. Please see my comments. I would recommend not publishing at this stage.

Comments:

"...these were essential for determination of the high-resolution structure of the two complexes "(line 85). Please specify why those mutations are important for this study. Does it produce some kind of rigidity within the protein? Or if it has already been reported earlier, please provide the references.

Fig. 1a: Please make the colors visible, it is very difficult at this stage to differentiate.

3. Please specify whether the energy filter was also used for the K3 Summit detector or not (line 149). It is very crucial to mention this here.

4. Cryo-EM image processing:

Please specify the max. and min. diameters for the "LoG filter" used for particle picking (line 168) along with the probable diameter of the particles.

For the Ab-initio reconstruction in cryosparc 2.15. (line 174), was any kind of symmetry applied? What is the value of class similarity used for? This value is crucial for the differentiation of various classes due to the conformational flexibility of protein or complex and sometimes fails to differentiate the conformational changes due to protein dynamics or complex formation.

I would rather suggest doing a refinement job using particles extracted from 2D classification, and then doing a 3D classification without alignment (authors can also use the mask) in relion and checking the different classes authors will have.

Which software did the authors use to do "homogeneous refinement" (like 182), cryosparc 2.15? Please mention.

Please specify the equation used for the "Custom python scripts" or if it was already available then provide references.

5. Model building:

Clash scores (from the Supplementary Table 1) are high and need additional refinements. What value of the nonbonded weight parameter was used? Authors need to consider changing this value or applying the occupancy strategy to improve the clash score.

4. MD simulation:

Why did the authors use 150 mM of Na-ion (line 322) instead of using 100 mM NaCl for sample preparation or buffer exchange? A 50 mM difference in salt could create a huge impact on Coulombic potential energy.

For the NVT simulation what thermostat was applied?

Please specify the type of box and size of the box used for the simulation.

Please provide one figure that contains the comparison of RMSD from three different sets of MD simulations.

There was no indication of which water model was used in this work, please specify.

6. HDX

5% Glycerol was used for the experiment, as it can increase the viscosity of the solution, so the HD exchange rate may change from samples prepared in buffer without glycerol. Did the authors consider that fact?

We thank all the reviewers for the constructive suggestions and comments. Our point-by-point responses to each reviewer's comments are listed below in blue text.

Reviewer #1 (Remarks to the Author):

Xi Lin et al. report the active structures of GPR21, which is one of the class A GPCRs, in complex with heterotrimeric miniGs and miniG15 proteins, respectively. They found an agonist-like motif in extracellular loop 2 that occupies the orthosteric pocket and promotes receptor activation. They also found that the flexible cytoplasmic portion of transmembrane helix 6 (TM6) adopts little or undetectable outward movement, which is different from other GPCRs such as β -2AR. In addition, they carried out HDX-MS experiments and molecular dynamics (MD) simulations to analyze the dynamic properties of TM6 and its neighboring residues. They executed their MD simulations of GPR21 without the G proteins, starting from three different modeled structures that modeled the missing residues of their EM structure. Their simulation length of 2 μ s and the number of parallel trajectories per system (three independent runs per modeled structure) are enough. Their simulation data exhibited local flexibility of TM6 and its neighbors, which agrees with the results of their functional and HDX-MS experiments. Therefore, their MD data could be published without any further revisions. Finally, since the structure of GPR21, which could be a potential novel drug target for diabetes, has not been reported before, their work would be worthwhile for publication in Nature Communications. However, there are some minor issues that should be revised in their manuscript before publication.

Response: We thank the reviewer's positive comments on our study.

Minor issues:

(i) I think if there was a sequence alignment comparing multiple GPCRs (beta2-AR etc.), especially TM6 and its neighbors, their results would become more understandable and convincing.

Response: Thanks for the suggestion! We have added sequence alignment of GPR21, GPR52 with representative class-A GPCRs (including β 2-AR) at ICL3-TM6 region in **Supplementary Figure 8** in the revised manuscript.

(ii) Upon paper acceptance, they could submit their MD data to an archive, such as GPCRmd (<https://submission.gpcrmd.org/home/>) or BSM-Arc (<https://bsma.pdbj.org/>). Considering the commitment of this journal to sharing data associated with its original publications, the authors should consider doing so.

Response: We have submitted our MD trajectories to **BSM-Arc** (DOI: 10.51093/bsm-00037) and the data will be released upon paper acceptance.

(iii) There are some typos such as 'A side pocket that may by employed ...' and 'induced-insulin resistance6. and its ...', so they should carefully revise their manuscript.

Response: Thank you for the comment! We revised the manuscript carefully and

corrected the typos.

Reviewer #2 (Remarks to the Author):

The work presented by Professor Xu and colleagues is very interesting and adds value to the field. The experiments are well planned and executed, and the results are explained clearly.

Response: We thank the reviewer's positive comments. As will be found below, we have incorporated the analysis as requested and deposited HDX-MS raw data to a public repository (IPX0005456000). We have provided a summary in the **Supplementary Table 7** in the revised manuscript as well.

I do have a few questions and suggestions for the authors:

1. The H/DX-MS data for TM6-ICL3 of all the three states of the protein is nicely depicted in supplementary figure 7. Interestingly, the authors showed that replacing Q235-V253 of apo-GPR21 (m5) with E212-L241 leads to the complete disappearance of the slowly exchanging population. I request the authors to provide the relative ratio between the two populations (Left and Right) for apo-GPR21 (m5) and apo-GPR21(m5)_P6.29A and the SD.

Response: We thank the reviewer for this comment. We have used HX express 2 software to analysis the EX1 kinetics of HDX data, which is summarized in the table below (also in the **Supplementary Table 5** in the revised manuscript).

Supplementary Table 5. Relative ratio between the two populations (Left and Right) for apo-GPR21 (m5) and apo-GPR21(m5)_P6.29A

apo GPR21 (m5)	10s	60s	300s	900s
bimodal(left) population %	26.14±2.3	26.14±2.3	32.69±1.88	8.92±2.26
bimodal(right) population %	74.46±4.06	73.86±2.3	67.31±1.88	91.08±2.26
apo GPR21(m5)_P6.29A	10s	60s	300s	900s
bimodal (left) population %	33.15±2.74	31.99±2.16	32.75±1.64	34.09±3.59
bimodal (right) population %	66.85±2.74	68.01±2.16	67.25±1.64	65.91±3.59

2. The legend of the Supp. Fig. 7a says mass spectra of 0s and 60s are shown, but there are also 10s, 300s, and 900s. Please rectify that.

Response: We apologize for the confusion. We have revised the legend for **Supp. Fig. 9a** (originally Supp. Fig. 7a) by adding "The values listed under each HDX experiments demonstrate of all exchange time points (i.e., 0s, 10s, 60s, 300s, 900s)."

3. Are the mean centroid values for the two populations of apo-GPR21 and apo-GPR21(m5)_P6.29A different? Please provide a table with the centroid values of each population and SD.

Response: We thank the reviewer for this comment. The mean centroid values are different between *apo* GPR21 and *apo* GPR21(m5)_P6.29A due to the fact that we detected different peptides for EX1 analysis.

protein	sequence	charge	monoisotopic mass (m/z)
GPR21	SSQSGETGEVQACPKRYAMVL	3	786.035
GPR21(m5)_P6.29A	SSQSGETGEVQACADKRYAMVL	3	777.363

We summarized the mean centroid values as well as the centroid values of left and right peak population as shown in the table below (also in the **Supplementary Table 6** in the revised manuscript).

Supplementary Table 6. Centroid value summary

apo GPR21 (m5)	10s	60s	300s	900s	Dmax
mean peak centroid	787.47±0.03	787.40±0.07	787.68±0.15	787.85±0.02	789.05±0.06
left peak centroid	786.58±0.01	786.50±0.03	786.68±0.07	786.66±0.10	NA
right peak centroid	787.74±0.05	787.75±0.09	788.24±0.10	787.97±0.06	NA
apo GPR21(m5)_P6.29A	10s	60s	300s	900s	Dmax
mean peak centroid	778.43±0.02	778.48±0.02	778.49±0.01	778.42±0.06	778.77±0.01
left peak centroid	777.89±0.01	777.89±0.02	777.87±0.01	777.88±0.01	NA
right peak centroid	778.72±0.03	778.75±0.01	778.75±0.01	778.71±0.03	NA

4. The authors have not explained in their methods how the carryover effect was minimized. Could the bimodal distribution be because of the carryover effect?

5. I understand that the average back exchange for H/DX-MS experiments can range between 25-30%, and based on that, the authors have corrected the uptake in the HDX workbench software. But, the BE % varies from peptide to peptide based on the length and accessibility. Therefore, I request the authors to perform a fully deuterated control experiment to show that the particular region TM6-ICL3 does shift to the maximally exchanged population with time.

Response for comment 4 and 5:

We thank the reviewer for this comment. We have incorporated the analysis of a fully deuterated control and updated the **Supplementary Fig. 9**, wherein the bimodal distribution disappeared. Therefore, the observed the EX1 distribution is not due to the carryover effect in our HDX platform. We have updated the relevant text in the figure legend and method.

6. Are there any other overlapping peptides from this region? It would be worth looking at those since the authors hypothesize that P6.29 plays an important role.

Response: We thank the reviewer for this comment. This is the only peptide we found in that region.

7. The authors explain data rendering in their methodology, but there are no results

showing this. Since the main data (peptide from TM6-ICL3) is showing a bimodal distribution, this would not be accurate. I suggest the authors remove this paragraph from their methodology.

Response: We thank the reviewer for this comment. We have removed the paragraph of data rendering.

I recommend the publication of the manuscript once the above concerns are satisfied.

Reviewer #3 (Remarks to the Author):

This is an interesting and solid paper which provides the first structures of the enigmatic orphan GPCR GPR21. Given the accelerating interest in orphan and understudied GPCRs this work will be important for a large number of biochemists, pharmacologists and cell biologists.

Response: We thank the reviewer's positive comments on our study.

SPECIFIC COMMENTS:

1. The lower resolution structures are incomplete per my examination of the maps. A careful examination of the G15 structure, for instance, disclosed a large number of residues whose side-chains were not easily discernable in the maps. As well a number of discontinuities in the low resolution maps was evident. By contrast the best map (which is the one discussed in the paper) was fine and the model accurately fits the map. I would recommend the authors either delete the lower resolution maps or clearly state in the results and discussion that the map-model correspondence is incomplete and indicate in a separate figure where side-chains and backbone carbons have been modeled in without correspondence with the map.

Response: As suggested by the reviewer, we deleted the side chains of residues in the lower resolution regions for the two GPR21(wt) maps where the resolvability is not sufficiently high. We have uploaded the updated models in the PDB database under accession numbers (**PDB: 8HJ1 for GPR21(wt)-Gs and 8HJ2 for GPR21(wt)-G15**). **On page 3 (line 83-85)** in the revised manuscript, we stated that "Some side chains were trimmed in the two models during model building since the map-model correspondence would be incomplete at some regions owing to the relatively low resolution". We also remade the corresponding figures of GPR21(wt)-Gs and GPR21(wt)-G15 complexes in **Supplementary Figure 1f and 2f**. The side-chains and backbone carbons of GPR21(wt)-G protein complexes are now well modeled in correspondence with the map.

2. For the estimates of constitutive activity the authors should as a control also have results with a GPCR with low levels of constitutive activity for comparison.

Response: Thanks for the comment! We added the control in the results shown in **Figs. 2a, c and Fig. 4c** in the revised manuscript. DRD2 was used as negative control for Gs

signaling and P2Y12 for G15 signaling.

3. Given the authors do not have an inactive state structure to compare with the active state structures, the section on mechanisms of activation is premature and should be eliminated or shortened and also the authors need to indicate that they are not making these inferences based on any real knowledge of the inactive  active state transitions for GPR21

Response: We agreed with this suggestion and used “**active conformation of GPR21**” to replace the original “**activation mechanism**” for the subtitle **on Page 6** in the revised manuscript. We also added the claim that “there is no inactive state GPR21 structure available to compare with the active state structures” **on Page 6 (line 161)** in the revised manuscript. In this part (active conformation of GPR21), we weakened the statement of activation mechanism of GPR21, instead, we mainly emphasized that the active conformation of GPR21 is unique. At the same time, we used an inactive model of GPR21 from GPCRDB, predicted by alphafold, to compare with our G protein-coupled GPR21 structures and observed conformational changes consistent with the comparison to the inactive-state GPR52 structure as we have previously shown, thus further supporting our conclusion: “GPR21 adopts the least conformational changes in active conformation”. The corresponding figure is now shown in **Supplementary Fig. 6** in the revised manuscript.

Reviewer #4 (Remarks to the Author):

In this manuscript, the authors investigated and compared the structures and dynamics of GPR21 and its mutant. The authors applied single particle cryo-EM, HDX, and other experiments to report the role of TM6 and a conserved ECL2 conformation in GPR21, essential for their high basal activity. The application of MD simulation also predicted the dynamic nature of TM6. This work is important in the field of drug development and represents a good correlation of experimental results with theoretical predictions. However, when I went through it in detail, I found that it needs a lot of explanation to be a good fit for this journal. Please see my comments. I would recommend not publishing at this stage.

Response: We thank the reviewer for the constructive comments. Over the revision stage, we have carried out new experiments and analysis to address these concerns.

Comments:

“...these were essential for determination of the high-resolution structure of the two complexes ”(line 85). Please specify why those mutations are important for this study. Does it produce some kind of rigidity within the protein? Or if it has already been reported earlier, please provide the references.

Response: Thanks for the comment! These mutations in GPR21 were designed based on sequence homology with GPR52. Indeed, the corresponding mutations were employed in the GPR52 constructs used for successful structural determination (PDB:

6LI0, 6LI1). In GPR52, these mutations were originally designed by our collaborators Vsevolod Katritch and Petr Popov using computational approach (CompoMug) for effective prediction of stabilizing mutations in GPCRs (Elife. 2018 Jun 21;7:e34729. doi: 10.7554/eLife.34729.). Then we screened these mutations one by one and combined several good mutations leading to the optimized GPR52 constructs. Given the high sequence homology between GPR52 and GPR21, we used the same mutations for GPR21 to stabilize the protein amenable for structural determination. To clarify this, we added the explanations and cited corresponding papers **on Page 3 (Line 88&89)** in the revised manuscript.

Fig. 1a: Please make the colors visible, it is very difficult at this stage to differentiate.

Response: Thanks for the comment! We revised the **Fig. 1a and Fig. 1b** to make the colors distinguishable in the revised manuscript.

3. Please specify whether the energy filter was also used for the K3 Summit detector or not (line 149). It is very crucial to mention this here.

Response: Thanks for the comment! We stated in the **Methods (Line148)** in the revised manuscript: all the datasets were collected on a Titan Krios 300 kV electron microscope (Thermo Fisher Scientifics, USA) equipped with a GIF Quantum energy filter (20 eV energy slit width, Gatan Inc., USA).

4. Cryo-EM image processing:

Please specify the max. and min. diameters for the “LoG filter” used for particle picking (line 168) along with the probable diameter of the particles.

Response: We specified it in the revised manuscript in the **Methods (Line 171)**: “Min: 60Å, max: 150Å”. The probable diameter of the particles is in 80-130Å range.

For the Ab-initio reconstruction in cryosparc 2.15. (line 174), was any kind of symmetry applied? What is the value of class similarity used for? This value is crucial for the differentiation of various classes due to the conformational flexibility of protein or complex and sometimes fails to differentiate the conformational changes due to protein dynamics or complex formation.

Response: No symmetry was applied during the ab-initio reconstruction by cryosparc v2.15. During this step, the class similarity value is 0.1 (the default value). During the subsequent Relion refinement, we chose one out of the five initial models which projections best match the 2D class averages for 3D classification. Relion will add certain levels of random noise to the initial model, and use them as the initial seeds for 3D classification step. While during the iterative classification process, particles with different conformations are likely to be classified into different groups, and particles with homogeneous conformation are likely to be grouped together. After convergence, the 3D refined maps of different classes will show the conformational differences among different groups.

I would rather suggest doing a refinement job using particles extracted from 2D

classification, and then doing a 3D classification without alignment (authors can also use the mask) in Relion and checking the different classes authors will have.

Response: Thank you for the suggestions. We have performed the 3D classification without alignment for all the datasets. For the GPR21(wt)-G15 dataset, the 3D classification results more or less resemble the ones we have obtained with alignment. Unfortunately, other datasets failed to be classified into meaningful subclasses. The results are shown in Figure below.

Which software did the authors use to do “homogeneous refinement” (like 182), cryosparc 2.15? Please mention.

Response: Thank you for the comment. In **the Methods section line 182**, we clarified the statement as this: “Subsequent 3D refinements, **including heterogeneous 3D refinements, homogeneous refinements, and auto refinement of all the datasets, were all performed by Relion3.1**”.

Please specify the equation used for the “Custom python scripts” or if it was already available then provide references.

Response: The script has not been published, and the purpose of this custom script is to eliminate “bad” particles that cannot be aligned to the consistent Euler angles during refinement. Implementing the script can only marginally improve the resolution by 0.1~0.2Å, or even has no improvements at all in some cases. Thus, the script is not essential to obtain high resolution structures. In this script, we obtained the Euler angles of each particles in 8 additional refinement iterations after convergence, and calculate the Euler angle differences between the adjacent iterations. (Let us assume that the data converges after iter 20, we calculate the Euler angle difference between the iter20 and 21, between iter 21 and 22, etc, and between iter 26 and 27. Then we count how many times the Euler angle differences are larger than 9 degrees. If the count is higher than 3.5, we regard the particle as “non-converged particle” and eliminate it in subsequent

refinement processes.)

The equation to convert from the polar coordinate to the Euclidean coordinate is as follows:

$a = \text{phi}, b = \text{theta}, c = \text{psi}.$

$x1 = \cos(a) \cdot \cos(b) \cdot \cos(c) - \sin(a) \cdot \sin(c) - \cos(c) \cdot \sin(a) - \cos(a) \cdot \cos(b) \cdot \sin(c) + \cos(a) \cdot \sin(b)$

$y1 = \cos(a) \cdot \sin(c) + \cos(b) \cdot \cos(c) \cdot \sin(a) + \cos(a) \cdot \cos(c) - \cos(b) \cdot \sin(a) \cdot \sin(c) + \sin(a) \cdot \sin(b)$

$z1 = \sin(b) \cdot \sin(c) + \cos(b) - \cos(c) \cdot \sin(b)$

Then we calculate the Euclidean distances between adjacent iterations and use the following equation to determine if the value is larger than 9 degree (approximate value) :

$$E_{\text{dist}} > 9/360 * 2\pi * \text{sqrt}(3)$$

This custom python script was also used for other GPCR structure determination published by our group, for example the FZD7-Gs complex structure (Cell Research, 2021 Dec;31(12):1311-1314). We cited this paper when we mention “custom python script” in the revised manuscript (in **Methods section in line 188**).

5. Model building:

Clash scores (from the Supplementary Table 1) are high and need additional refinements. What value of the nonbonded weight parameter was used? Authors need to consider changing this value or applying the occupancy strategy to improve the clash score.

Response: Thank you for the suggestions. Initially the refinement was performed by the default nonbonded weight (100). After changing to higher values, we did improve the clash scores. We have updated the refinement stats in the **Supplementary Table 1** in the revised manuscript.

4. MD simulation: Why did the authors use 150 mM of Na-ion (line 322) instead of using 100 mM NaCl for sample preparation or buffer exchange? A 50 mM difference in salt could create a huge impact on Coulombic potential energy. For the NVT simulation what thermostat was applied? Please specify the type of box and size of the box used for the simulation. Please provide one figure that contains the comparison of RMSD from three different sets of MD simulations. There was no indication of which water model was used in this work, please specify.

Response: Thanks for the suggestions! 150 mM Na-ion is the most commonly used condition in MD simulations of GPCRs (e.g. in doi: 10.1038/s41589-021-00890-8) because it is the physiological concentration of Na-ion in blood and interstitial fluids. We used 150 mM Na-ion because we aimed to use MD simulations to explore the dynamics of GPR21 in physiological condition. We have added note of physiological concentration in **Methods section in line 328**.

For NVT simulation, we used Berendsen thermostat. We have added notes of thermostat and barostat in **Methods section in line 334-339**.

We have added box type, box size, and water model in **Methods section in line 325**. We have added figures of RMSD in MD simulations in **Supplementary Figure 10c**.

6. HDX: 5% Glycerol was used for the experiment, as it can increase the viscosity of the solution, so the HD exchange rate may change from samples prepared in buffer without glycerol. Did the authors consider that fact?

Response: Thank you for this comment. We have incorporated the analysis of a fully deuterated control and updated the **Supplementary Fig. 9**, wherein the bimodal distribution disappeared. Therefore, the showed that the EX1 distribution is not due to the carryover effect or 5% Glycerol within the sample buffer. We have updated the relevant text in the **figure legend** and in **Methods section** as well in the revised manuscript.

REVIEWERS' COMMENTS

Reviewer #2 (Remarks to the Author):

I thank the authors for satisfactorily answering my queries and adjusting the manuscript accordingly. However, upon review of the supplementary figure 9, I opine that the exchange heatmap can be removed and enlarge the spectra for a better figure resolution.

Reviewer #3 (Remarks to the Author):

The authors have addressed my concerns fully.

We thank the editors and all the reviewers for the constructive suggestions and comments. Our response to reviewer's comment is listed below in blue text.

Reviewer #2 (Remarks to the Author):

I thank the authors for satisfactorily answering my queries and adjusting the manuscript accordingly. However, upon review of the supplementary figure 9, I opine that the exchange heatmap can be removed and enlarge the spectra for a better figure resolution.

Response: Thanks for the comment. We removed the exchange heatmap for a better figure resolution of spectra in the Supplementary Figure 9 in the revised manuscript.